# Expanding Scaling Boundary of Compositional Text-to-Image Generation via Composition Curriculum

## Abstract

Text-to-Image (T2I) generation has long been an open problem, with compositional synthesis remaining particularly challenging. This task requires accurate rendering of complex scenes containing multiple objects that exhibit diverse attributes as well as intricate spatial and semantic relationships, demanding both precise object placement and coherent inter-object interactions. In this paper, we propose a novel compositional curriculum reinforcement learning framework for T2I generation, named CompGen, to address compositional weaknesses in T2I models. Specifically, we leverage scene graphs and introduce a novel difficulty criterion along with a corresponding adaptive Markov Chain Monte Carlo graph sampling algorithm. Using this difficulty-aware approach, we generate training datasets for Group Relative Policy Optimization (GRPO) comprising prompts and question-answer pairs with varying complexity levels. We demonstrate that different training schedulers yield distinct scaling curves for GRPO, with data distributions following easy-to-hard progression or Gaussian sampling strategies producing superior scaling performance than random. Our extensive experiments demonstrate that CompGen significantly strengthens compositional generation capabilities for both diffusion and auto-regressive T2I models, which highlights its effectiveness in enhancing the compositional understanding of T2I generation systems.

## 1 Introduction

Text-to-Image (T2I) generation has achieved remarkable progress in synthesizing visually compelling content from textual descriptions (Ramesh et al., 2022; Saharia et al., 2022; Podell et al., 2023; Balaji et al., 2022). Despite these advances, current T2I models face significant limitations in compositional synthesis, particularly in accurately rendering complex scenes containing multiple objects with diverse attributes and intricate spatial relationships (Liu et al., 2022; Nie et al., 2024). Thus, compositional T2I generation with complex numerical instructions (Huang et al., 2025; Hu et al., 2024) is still an open problem. To cope with this challenge, plenty of literature focuses on developing new network architectures such as attention models (Chefer et al., 2023; Rassin et al., 2023; Meral et al., 2024; Kim et al., 2023), or introducing intermediate structures like object layout (Chen et al., 2024; Dahary et al., 2024; Wang et al., 2024c). Unlike recent methods that require synthesized ground-truth images (Gao et al., 2024; Sun et al., 2023a) or intermediate skeletons (Nie et al., 2024) for supervised fine-tuning, our approach takes a data-centric perspective and enhances compositional generalization solely through textual prompts, applying reinforcement learning (RL) without requiring ground-truth image outputs. However, large-scale compositional RL training can be highly unstable due to the heterogeneous capability requirements of compositional T2I generation tasks, which encompass object existence, attribute existence, relational understanding, and numerical counting.

To address this challenge, we propose **CompGen**, a novel compositional curriculum reinforcement learning framework for text-to-image (T2I) generation. CompGen draws inspiration from human cognitive development, which follows a curriculum learning progression: first mastering the recognition and generation of individual objects and their attributes within simple relational contexts, then gradually learning to understand and create complex multi-object compositions involving multiple relations. Specifically, our approach leverages scene graphs (Krishna et al., 2017) as a compositional

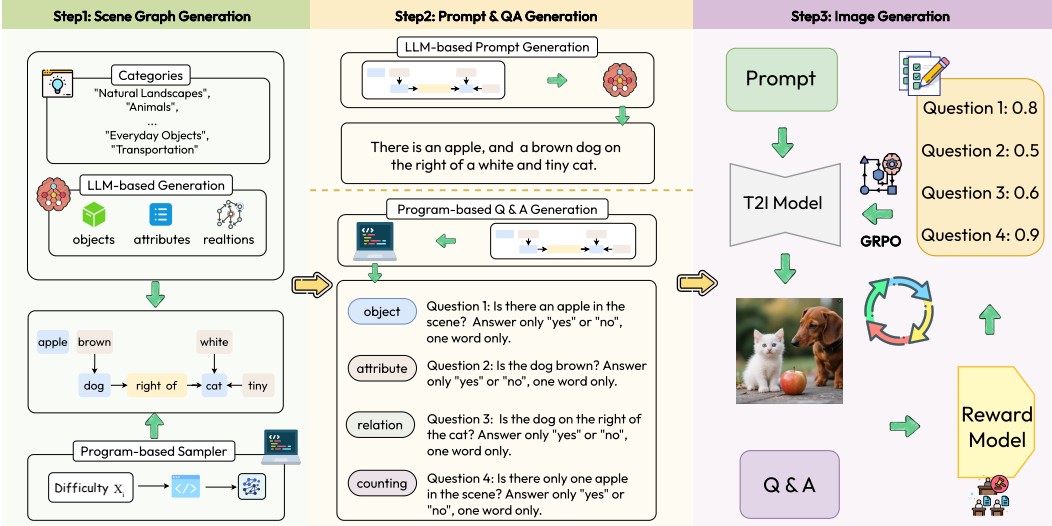

Figure 1: Overview of our CompGen framework, which is incentivized to construct a curriculum through end-to-end reinforcement learning without requiring ground-truth images.

representation of visual scenes to systematically generate training data with controllable complexity. First, we employ LLMs to generate diverse scene graph assets, where each scene graph encodes a structured representation of objects, their attributes, and relational dependencies. Second, we introduce a novel difficulty criterion that quantifies compositional complexity based on scene graph structural properties, including entity count, attribute diversity, and relational interconnectedness. Third, leveraging this difficulty metric, we develop an adaptive Markov Chain Monte Carlo (MCMC) sampling algorithm (Geyer, 1992) to systematically generate scene graphs at targeted complexity levels, thereby enabling precise curriculum control throughout the training process. Finally, for each sampled scene graph, we synthesize corresponding text prompts for image generation and construct comprehensive visual question-answer pairs for assessments, namely object existence, object counting, attribute recognition, and relational understanding. These question-answer pairs subsequently serve as reward metrics within our RL framework, guiding the model toward improved compositional reasoning performance.

Our extensive experiments demonstrate that CompGen significantly strengthens compositional generation capabilities across both diffusion and autoregressive T2I architectures. On five established compositional generation benchmarks — GenEval (Ghosh et al., 2023), DPG (Hu et al., 2024), TIFA (Hu et al., 2023), T2I-Bench (Huang et al., 2025), and DSG (Cho et al., 2023) — our approach consistently outperforms baseline models, achieving an average improvement of 12.52% when applied to Stable-Diffusion-1.5 and 3.44% when applied to LlamaGen. Furthermore, our analysis reveals that different training schedulers produce distinct scaling behaviors in GRPO training, with easy-to-hard progression and Gaussian sampling strategies demonstrating superior compositional performance and extending the scaling boundaries compared to random sampling approaches.

## 2 RELATED WORK

Large text-to-image (T2I) generative models have attracted considerable attention in recent years and can be broadly categorized into two main families: diffusion-based models (Ramesh et al., 2022; Rombach et al., 2022; Podell et al., 2023; Balaji et al., 2022) and auto-regressive models (Ramesh et al., 2021; Yu et al., 2022; Chang et al., 2023). Beyond purely improving visual quality, recent investigations (Wang et al., 2024a;b) have focused on enhancing prompt-following capabilities, particularly for compositional prompts. However, existing T2I models often struggle with compositional understanding, leading to issues such as object omission and incorrect attribute binding (Okawa et al., 2023; Huang et al., 2025). To address these limitations, recent efforts to improve compositional alignment can be grouped into three main categories. **Attention-based methods** modify attention maps within the UNet architecture to enforce object presence and spatial separation (Chefer et al., 2023; Rassin et al., 2023; Feng et al., 2022; Meral et al., 2024; Kim et al., 2023). For instance, DenseDiffusion (Kim et al., 2023) adjusts attention scores in both cross-attention and self-attention

layers to ensure object features align with specified image regions, while CONFORM (Meral et al., 2024) strengthens associations between relevant objects and attributes through contrastive objectives. However, these approaches face scalability and computational efficiency limitations as they operate only during inference. **Planning-based methods** utilize intermediate structures such as object layouts — either manually defined (Chen et al., 2024; Dahary et al., 2024; Wang et al., 2024c) or generated by large language models (Gani et al., 2023; Lian et al., 2023) — to guide image synthesis. Some approaches incorporate additional modules like visual question-answering models or captioning models for refinement (Yang et al., 2024; Wu et al., 2024; Wen et al., 2023); however, these additions increase inference costs and may still suffer from incorrect attribute bindings due to inherent model limitations. **Learning-based methods** focus on training-time improvements, including fine-tuning diffusion models with vision-language supervision (Wang et al., 2024c; 2022; Fan et al., 2023) or employing reinforcement learning techniques (Fan et al., 2023; Black et al., 2023). Caption-guided optimization represents another promising direction in this category (Fang et al., 2023; Ma et al., 2024). Unlike previous approaches that require additional inputs or architectural modifications, our method adopts a data-centric strategy to enhance compositional generalization through textual prompts and targeted training alone, without increasing inference costs or altering model architecture.

## 3 PRELIMINARY: SCENE-GRAPH AS A DIFFICULTY MEASURER

The core principle of CompGen is to progressively develop compositional generation capabilities through curriculum learning, advancing from simple to complex samples. The central challenge in this approach lies in establishing a principled framework for defining and quantifying the difficulty of compositional samples. Following the formalized image representation framework introduced by Krishna et al. (2017), scene graphs provide a structured approach to capturing compositional complexity. We therefore propose to measure sample difficulty through the compositional complexity inherent in scene graph structures. Formally, we provide the definition of difficulty based on scene graphs as follows:

**Definition 1** (Scene Graph Formulated Difficulty). *Given a scene graph $\mathcal{G} = (\mathcal{O}, \mathcal{A}, \mathcal{R})$ where $\mathcal{O}$ is the set of objects, $\mathcal{A}$ is the set of attributes associated with the objects, and $\mathcal{R}$ is the set of relations, we measure the difficulty of $\mathcal{G}$ as:*

$$\text{Diff}(\mathcal{G}) = \|\mathcal{O}\| \cdot \max\left(1, \frac{\|\mathcal{A}\|}{\|\mathcal{O}\|}\right) \cdot \max\left(1, \frac{\|\mathcal{R}\|}{\|\mathcal{O}\|}\right), \tag{1}$$

*where, the total number of objects is $\|\mathcal{O}\|$, the average attribute density is $\|\mathcal{A}\|/\|\mathcal{O}\|$, and the average relational connectivity is $\|\mathcal{R}\|/\|\mathcal{O}\|$ per object, where relations are treated as directed edges in the scene graph representation.*

Eq. (1) demonstrates that the computational difficulty of $\mathcal{G}$ is determined by three key factors identified by the total number of objects, the average attribute density and the average relational connectivity. Consequently, the overall difficulty exhibits a positive correlation with the intrinsic complexity of the scene graph, reflecting both its structural density and semantic richness.

## 4 COMPGEN: RL TRAINING WITH COMPOSITION CURRICULUM

Given the difficulty measure introduced in Definition 1, we can construct a mapping function $\{\mathcal{G}\} \to \mathbb{R}^+$ over the space of scene graphs $\mathcal{G}$. Building upon this foundation, we formulalized our problem as follows:

**Definition 2** (Compositional Text-to-Image Generation with Difficulty Constraint RL). *Given a target difficulty range $[\text{Diff}_{\min}, \text{Diff}_{\max}]$ where $0 < \text{Diff}_{\min} \leq \text{Diff}_{\max}$, our objective is to generate a pair $(T, R)$ consisting of: (i) An input text prompt $T$ that describes a visual scene with compositional complexity and (ii) A reward function $R : \mathcal{I} \to \mathbb{R}$ that evaluates generated images $I \in \mathcal{I}$.*

*Therefore, our methodology follows a structured pipeline: we first construct a scene graph $\mathcal{G}$ satisfying the difficulty constraint $\text{Diff}_{\min} \leq \text{Diff}(\mathcal{G}) \leq \text{Diff}_{\max}$, then derive the corresponding input text $T$ from this graph. These text prompts are processed by a T2I model to generate images $I$, which are subsequently evaluated by the reward function $R$. The resulting rewards enable optimization of the T2I model within a RL framework, ensuring adherence to the specified difficulty constraints.*

Based on the problem formulation above, our CompGen framework comprises three key components: (i) scene graph generation under difficulty constraints, (ii) input text derivation from scene graphs, and (iii) reward function design for reinforcement learning optimization. Each component is thoroughly examined in the following subsections.

## 4.1 Scene Graph Generation via Adaptive Markov Chain Monte Carlo

Given the target difficulty bounds $\text{Diff}_{\min}$ and $\text{Diff}_{\max}$, we define the constrained sampling problem as the task of generating scene graphs $\mathcal{G}$ such that $\text{Diff}_{\min} \leq \text{Diff}(\mathcal{G}) \leq \text{Diff}_{\max}$. This problem presents significant computational challenges, requiring efficient exploration of the high-dimensional graph space while simultaneously maintaining detailed balance for unbiased sampling, preserving structural validity of scene graphs, and enabling adaptive exploration under the imposed difficulty constraints.

To overcome these challenges, we develop a Markov Chain Monte Carlo (MCMC) sampling strategy (Brooks, 1998) that efficiently targets specific difficulty ranges. This approach enables sampling from a distribution concentrated on graphs with desired difficulty levels while avoiding computationally prohibitive explicit enumeration. Our method employs a set of four reversible graph transformation operations $\mathcal{T} = \{t_i\}_{i=1}^4$, which correspond to node addition, node removal, edge addition, and edge removal operations. These transformations are designed to ensure both ergodicity and detailed balance—fundamental requirements for unbiased MCMC sampling. We further define $\mathcal{T}_{\text{valid}} \subseteq \mathcal{T}$ as the subset of transformations that preserve essential graph properties, including connectivity and structural validity. The sampling procedure is directed by an energy function (i.e., $\text{Energy}(\mathcal{G})$) that quantifies the deviation from the target difficulty range:

$$\text{Energy}(\mathcal{G}) := \Delta(\mathcal{G}) = \text{Dist}(\text{Diff}(\mathcal{G}), \text{Diff}_{\min}, \text{Diff}_{\max}) = \begin{cases} \text{Diff}_{\min} - \text{Diff}(\mathcal{G}), & \text{if } \text{Diff}(\mathcal{G}) < \text{Diff}_{\min} \\ \text{Diff}(\mathcal{G}) - \text{Diff}_{\max}, & \text{if } \text{Diff}(\mathcal{G}) > \text{Diff}_{\max} \\ 0, & \text{otherwise,} \end{cases}$$

(2)

where $\text{Dist}(\text{Diff}(G), [\text{Diff}_{\min}, \text{Diff}_{\max}])$ is the distance function that quantifies the deviation of a scalar $\text{Diff}(\mathcal{G})$ from the target interval $[\text{Diff}_{\min}, \text{Diff}_{\max}]$, with the specific computation given above. We then adopt the Metropolis-Hastings algorithm (Chib & Greenberg, 1995) with acceptance probability defined as:

$$\text{Acc}(\mathcal{G}'|\mathcal{G}) = \min\left(1, \frac{\pi(\mathcal{G}')q(\mathcal{G}|\mathcal{G}')}{\pi(\mathcal{G})q(\mathcal{G}'|\mathcal{G})}\right),$$

(3)

where $\pi(\mathcal{G}) \propto \exp(-\text{Energy}(\mathcal{G})/\tau)$ is the target distribution and $q(\mathcal{G}'|\mathcal{G})$ is the proposal distribution. Since all our graph operations are symmetric, the proposal ratio cancels, yielding:

$$\text{Acc}(\mathcal{G}'|\mathcal{G}) = \min\left(1, \exp\left(\frac{\Delta(\mathcal{G}) - \Delta(\mathcal{G}')}{\tau}\right)\right).$$

(4)

To balance exploration and exploitation, we employ a logarithmic cooling schedule $\tau = 1/\log(t+2)$, which allows broad early exploration and progressively tighter focus near the target range. To avoid stagnation in local minima, the sampler reinitializes after a fixed number of stalled iterations, reshuffling graph structures while preserving key properties such as node counts and edge density.

## 4.2 Reinforcement Learning Strategies with Composition Curriculum

Having obtained a scene graph $\mathcal{G}$ through the constrained sampling process according to the above subsection, we turn to the generation of input text prompts $T$ and the design of appropriate reward functions $R$.

**Input Text Generation.** CompGen transforms structured scene graphs into natural language prompts while preserving their original compositional difficulty. Given a scene graph $\mathcal{G} = (\mathcal{O}, \mathcal{A}, \mathcal{R})$ with associated difficulty $\text{Diff}(\mathcal{D}) \in [\text{Diff}_{\min}, \text{Diff}_{\max}]$, we employ constrained Large Language Model (LLM)-based generation with strict constraints to produce descriptions that exactly match the graph's specifications. The generation process combines precise constraints with LLM capabilities to maintain strict fidelity to the source graph. This is achieved through three core mechanisms: mandatory inclusion of all objects and attributes ensures exact term matching, while structural and multi-stage

content validation preserve relationships and integrity. By combining the linguistic flexibility of LLMs with systematic safeguards such as controlled temperature scheduling and iterative validation, the process guarantees that the generated prompt fully reflects all elements of the original graph and preserves its difficulty. While implementing, we use Deepseek-V3 (Liu et al., 2024) as the LLM.

**Reward Generation.** For any text-to-image (T2I) model, generated images can be obtained from input text prompts. We evaluate the compositional accuracy of these images through structured question-answering pairs, which are systematically constructed from the sampled scene graphs. Specifically, we choose a programmatic approach (Gao et al., 2024) to generate precise and comprehensive binary questions that fully cover the scene graphs. Questions are generated directly from scene graphs $\mathcal{G} = (\mathcal{O}, \mathcal{A}, \mathcal{R})$. Since the scene graph encodes three types of information—objects, attributes, and relations—and the number of repeated objects is also a critical factor, we design four types of questions to cover all aspects: object verification questions ($Q_{\text{obj}}$) assessing presence of $o \in \mathcal{O}$, count questions ($Q_{\text{count}}$) verifying the number of occurrences of repeated objects in $\mathcal{O}$, attribute validation questions ($Q_{\text{attr}}$) checking $a \in \mathcal{A}$ for corresponding objects, and relation confirmation questions ($Q_{\text{rel}}$) verifying $r \in \mathcal{R}$ between object pairs.

Given the binary question-answer pairs we constructed, and inspired by VQAScore (Lin et al., 2024), we adopt a multimodal large language model (MLLM)'s predicted probability of answering "yes" as the initial fine-grained reward score. We take the average of the VQA scores for all questions corresponding to each image as the reward signal for reinforcement training, enabling the T2I model to perform policy updates. For ease of notation, we use $p_{\text{reward}}(\cdot)$ to denote the MLLM in use. While practical, we apply Llava-v1.6-13B (Liu et al., 2023), and we also evaluate different MLLMs in Section 5.3

**RL Training.** During training, we employ Group Relative Policy Optimization (GRPO) to optimize T2I model's compositional generation ability. For each text prompt $T$ sampled from dataset $\mathcal{D}$, GRPO generates $G$ distinct images $\{I^{(1)}, I^{(2)}, \ldots, I^{(G)}\}$ using the current policy $p_{\theta_{\text{old}}}$. The policy is optimized by maximizing:

$$\mathcal{J}_{\text{GRPO}}(\theta) = \mathbb{E}_{T \sim \mathcal{D}} \left[ \frac{1}{G} \sum_{i=1}^{G} \left( \min\left( \frac{\pi_\theta}{\pi_{\theta_{\text{old}}}} A_i, \quad \text{clip}\left( \frac{\pi_\theta}{\pi_{\theta_{\text{old}}}}, 1 - \epsilon, 1 + \epsilon \right) A_i \right) \right. \right.$$
$$\left. \left. - \beta \, \text{KL}\left( p_\theta(\cdot|T) \big\| p_{\text{ref}}(\cdot|T) \right) \right) \right], \tag{5}$$

where $\pi_\theta = p_\theta(I^{(i)}|T)$ denotes the probability of generating image $I^{(i)}$ given text prompt $T$ under the current policy, $\pi_{\theta_{\text{old}}} = p_{\theta_{\text{old}}}(I^{(i)}|T)$ is the probability under the old policy, $\epsilon$ and $\beta$ are hyperparameters for clipping and KL regularization respectively, and $p_{\text{ref}}$ is the reference policy.

For each generated image $I^{(i)}$, we evaluate its quality using binary question-answer pairs through our reward model as:

$$r_j^{(i)} = p_{\text{reward}}(\text{answer}_j | I^{(i)}, \text{question}_j), \tag{6}$$

where $r_j^{(i)} \in [0, 1]$ represents the reward model's confidence that image $I^{(i)}$ correctly answers binary question. The overall reward for image $I^{(i)}$ is computed as the average score across all the sampled questions: $r^{(i)} = \frac{1}{M} \sum_{j=1}^{M} r_j^{(i)}$, where $M$ is the number of samples. The advantages are normalized within each group of $G$ images:

$$A_i = \frac{r^{(i)} - \text{mean}(\{r^{(k)}\}_{k=1}^{G})}{\text{std}(\{r^{(k)}\}_{k=1}^{G})}, \tag{7}$$

where $\text{mean}(\{r^{(k)}\}_{k=1}^{G})$ and $\text{std}(\{r^{(k)}\}_{k=1}^{G})$ denote the mean and standard deviation of rewards within the group, ensuring zero-mean unit-variance normalization of advantages.

Table 1: Comparison of different T2I models on compositional generation benchmarks. Upward arrows indicate improvement direction; red numbers show improvement magnitude over baseline.

| Model | # Params | GenEval | DPG | TIFA | T2I-Bench | DSG | Avg. |
|---|---|---|---|---|---|---|---|
| *Diffusion* | | | | | | | |
| Stable-Diffusion-1.4 | 0.9B | 42.04% | 61.89% | 79.14% | 30.80% | 61.71% | 55.12% |
| Stable-Diffusion-1.5 | 0.9B | 42.08% | 62.24% | 78.67% | 29.94% | 61.57% | 54.90% |
| Stable-Diffusion-2.1 | 0.9B | 50.00% | 65.47% | 82.00% | 32.01% | 68.09% | 59.51% |
| Playground-V2 | 2.6B | 59.00% | 74.54% | 86.20% | 36.13% | 74.54% | 66.08% |
| Stable-Diffusion-XL | 2.6B | 55.87% | 74.65% | 83.50% | 31.30% | 83.40% | 65.74% |
| *Diffusion Transformer* | | | | | | | |
| PixArt-$\alpha$ | 0.6B | 48.00% | 71.11% | 82.90% | 41.17% | 71.11% | 62.85% |
| Lumina-Next | 1.7B | 46.00% | 75.66% | 79.98% | 34.57% | 70.61% | 61.36% |
| *AutoRegressive* | | | | | | | |
| Show-o | 1.3B | 56.00% | 67.27% | 86.28% | 29.00% | 77.00% | 63.11% |
| LlamaGen | 775M | 31.28% | 42.92% | 75.03% | 33.26% | 58.30% | 48.16% |
| Emu3 | 14B | 54.00% | 74.19% | 81.86% | 31.20% | 70.31% | 62.31% |
| minDALL-E | 1.3B | 23.00% | 55.23% | 79.40% | 18.98% | 45.63% | 44.45% |
| *Ours* | | | | | | | |
| Stable-Diffusion-1-5 w/ours | 0.9B | 53.88% (↑11.80) | 78.67% (↑16.43) | 85.71% (↑7.04) | 37.68% (↑7.74) | 77.16% (↑15.59) | 66.62% (↑11.72) |
| LlamaGen w/ours | 775M | 35.71% (↑4.43) | 48.67% (↑5.75) | 77.92% (↑2.89) | 36.14% (↑2.88) | 59.57% (↑1.27) | 51.60% (↑3.44) |

## 5 EXPERIMENT

### 5.1 EXPERIMENTAL SETUP

**Datasets and Benchmarks.** To thoroughly assess the model's compositional capability, we evaluate it on the following five compositional benchmarks: (i) **Geneval** (Ghosh et al., 2023), (ii) **T2I-CompBench** (Huang et al., 2025), (iii) **TIFA** (Hu et al., 2023), (iii) **DPG-Bench** (Hu et al., 2024), (iv) **DSG** (Cho et al., 2023). We provide a detailed description for each dataset in Appendix A.

**Baseline Models.** To validate the effectiveness of our proposed method, we conduct comprehensive comparisons against several state-of-the-art text-to-image generation models. These include diffusion models such as Stable Diffusion 1.4/2.1 (Rombach et al., 2022), Playground v2 (Li et al.), and Stable Diffusion XL (Podell et al., 2023); Diffusion Transformer-based models including PixArt-alpha (Chen et al., 2023) and LUMINA-NEXT (Zhuo et al., 2024); as well as auto-regressive T2I models such as Show-o (Xie et al., 2024), Emu3 (Sun et al., 2023b) and minDALL-E (Kim et al., 2021).

To generate high-quality training data for compositional RL, we employ the CompGen framework to construct 10K samples evenly distributed across difficulty levels ranging from 1 to 10. Each sample includes text prompts and corresponding question-answer pairs, with answers set to "yes" to enable a VQA model to evaluate whether the generated images contain the described content. To demonstrate the effectiveness and generalizability of our CompGen for enhancing compositional generation capabilities of T2I models, we select Stable-Diffusion v1.5 (Rombach et al., 2022) and LlamaGen (Sun et al., 2024), which cover two prominent T2I model architectures.

### 5.2 PERFORMANCE COMPARISONS AND ANALYSIS

**CompGen significantly enhances T2I compositional generation capabilities.** Our experimental results demonstrate that CompGen consistently delivers substantial improvements across all compositional generation benchmarks. Specifically, when applied to Stable Diffusion 1.5, CompGen achieves an average performance gain of 11.72%, elevating the baseline from 54.90% to 66.62%. This enhanced performance not only surpasses the original baseline but also outperforms several stronger contemporary models, including Stable Diffusion 2.1 (59.51%), PixArt-$\alpha$ (62.85%), and even competitive autoregressive models like Emu3 (62.31%). Notably, our enhanced SD-1.5 model with only 0.9B parameters achieves comparable or superior compositional generation performance to much larger models such as Playground-V2 (2.6B, 66.08%), demonstrating the efficiency and effectiveness of our approach in bridging the compositional generation gap without requiring substantial architectural modifications or parameter scaling.

Table 2: Performance changes of CompGen with various reward models.

| Model | Reward Model | GenEval | DPG | TIFA | T2I-Bench | DSG | Avg. |
|---|---|---|---|---|---|---|---|
| *Diffusion* | | | | | | | |
| Stable-Diffusion-1.5 | – | 42.08% | 62.24% | 78.67% | 29.94% | 61.57% | 54.10% |
| Stable-Diffusion-1.5 w/ VQAScore | LLaVA-v1.6-13B | 44.02% | 73.41% | 80.19% | 36.36% | 73.23% | 61.42% |
| | CLIP-FlanT5-XXL | 45.11% | 71.26% | 81.42% | 31.60% | 73.74% | 60.62% |
| CompGen (Stable-Diffusion-1.5) | InstructBLIP | 42.04% | 64.00% | 76.29% | **39.21%** | 64.58% | 57.22% |
| | LLaVA-v1.5-13B | 49.23% | 74.54% | 84.84% | 37.43% | 75.97% | 64.40% |
| | LLaVA-v1.6-13B | **53.88%** | **78.67%** | **85.71%** | 37.68% | **77.16%** | **66.62%** |
| *AutoRegressive* | | | | | | | |
| LlamaGen | – | 31.28% | 42.92% | 75.03% | 33.26% | 58.30% | 48.16% |
| LlamaGen w/ VQAScore | LLaVA-v1.6-13B | 33.15% | 44.51% | 75.94% | 34.28% | 59.46% | 49.47% |
| | CLIP-FlanT5-XXL | 32.47% | 44.57% | 75.98% | 33.71% | 58.60% | 49.07% |
| CompGen (LLamaGen) | InstructBLIP | 32.03% | 44.75% | 75.54% | 33.52% | 57.98% | 48.76% |
| | LLaVA-v1.5-13B | 31.09% | 45.72% | 75.96% | 31.28% | 57.54% | 48.32% |
| | LLaVA-v1.6-13B | **35.71%** | **48.67%** | **77.92%** | **36.14%** | **59.57%** | **51.60%** |

**Stronger base models exhibit greater improvements with CompGen.** Our analysis reveals a compelling correlation between base model capability and the magnitude of improvement achieved through CompGen training. The more capable Stable Diffusion 1.5 baseline (54.90%) experiences a substantial 11.72% improvement, with particularly pronounced gains in DPG (+16.43%) and DSG (+15.59%) benchmarks that specifically evaluate complex compositional reasoning. In contrast, the weaker LlamaGen baseline (48.16%) shows more modest improvements of 3.44% on average, with relatively smaller gains across individual benchmarks. This pattern suggests that CompGen's effectiveness is amplified when applied to models with stronger foundational generation capabilities.

**CompGen generalizes beyond compositionality without overfitting to training objectives.** As illustrated in Figure 3, our enhanced models maintain high-quality image generation across diverse scenarios while significantly improving compositional accuracy. The qualitative results demonstrate that CompGen effectively improves complex multi-object compositions, spatial relationships, and attribute bindings while preserving overall image fidelity and avoiding the introduction of artifacts.

## 5.3 Ablation Study

**Ablation on Reward Design.** We conduct an ablation study comparing our fine-grained reward design against VQAScore (Lin et al., 2024) to validate the effectiveness of our compositional reward formulation. While VQAScore employs a single binary question asking whether the prompt matches the image, our approach decomposes the evaluation into multiple fine-grained aspects including object existence, attribute presence, relational understanding, and counting accuracy. As shown in Table 2, our CompGen framework consistently outperforms VQAScore across both diffusion and autoregressive models. Specifically, when applied to Stable-Diffusion-1.5 with LLaVA-v1.6-13B as the reward model, CompGen achieves 66.62% average performance compared to 61.42% for VQAScore, representing a substantial 5.2 percentage point improvement. The gains are particularly pronounced on complex compositional benchmarks like GenEval (+9.86%) and DPG (+5.26%), demonstrating that fine-grained reward signals better capture the nuanced requirements of compositional generation. Interestingly, while autoregressive models show more modest improvements (51.60% vs 49.47% for LlamaGen), the consistent gains across different architectures validate the generalizability of our reward design.

**Ablation on Reward Model.** To investigate the impact of reward model capability on CompGen performance, we conduct ablation experiments across four different vision-language models: LLaVA-v1.6-13B (Liu et al., 2023), LLaVA-v1.5-13B (Liu et al., 2023), CLIP-FlanT5-XXL (Radford et al., 2021), and InstructBLIP-FlanT5-XXL (Dai et al., 2023). All experiments utilize our compositional fine-grained reward design with uniformly sampled 10K training data on both Stable-Diffusion-1.5 and LlamaGen base models. As shown in Table 2, the choice of reward model significantly influences

| CompGen | SD1.5 | SD2.1 | SDXL | Lumina-Next |
|---|---|---|---|---|

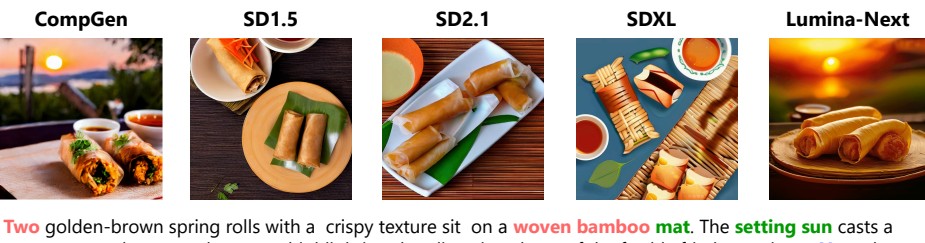

**Two** golden-brown spring rolls with a crispy texture sit on a **woven bamboo mat**. The **setting sun** casts a warm, orange hue over the scene, highlighting the glistening sheen of the freshly fried appetizers. **Near** the spring rolls, a small dish of **dipping sauce** reflects the sunset's glow, enticing one to indulge in the savory treat.

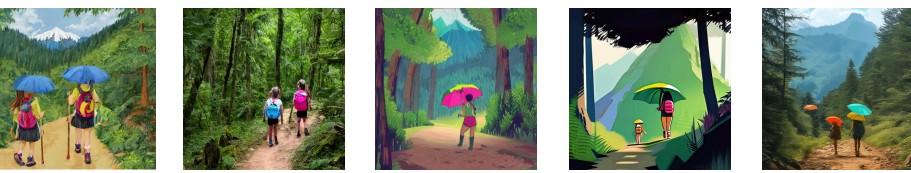

**Two** young girls are trekking on a dirt trail that meanders through a dense forest on a large, imposing mountain. The trees enveloping the path are lush and varying shades of green. Each girl is **holding** a **brightly colored umbrella** to shield themselves from the elements. The mountain's peak looms in the distance, partially obscured by the canopy of towering trees.

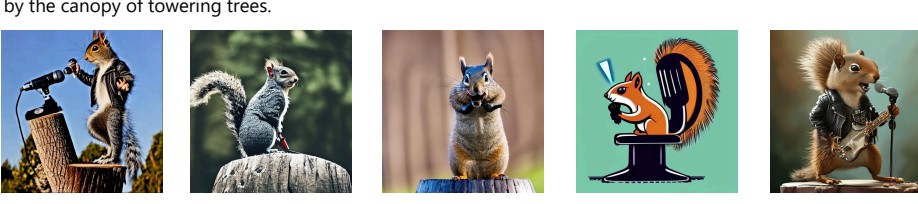

A **punk** rock squirrel in a studded leather **jacket** shouting into a **microphone** while **standing on** a **stump**.

Figure 2: Qualitative comparison of compositional generation across CompGen and other strong T2I models.

the effectiveness of our approach, with LLaVA-v1.6-13B consistently achieving the best performance across all benchmarks. Specifically, LLaVA-v1.6-13B outperforms the second-best LLaVA-v1.5-13B by 2.22% and 3.28% average improvement for diffusion and autoregressive models respectively, highlighting the importance of advanced vision-language understanding capabilities. Notably, the performance gap between different reward models is more pronounced in diffusion models (66.62% vs 57.22% for best vs worst) compared to autoregressive models (51.60% vs 48.32%), suggesting that diffusion models are more sensitive to reward signal quality. InstructBLIP shows surprisingly weaker performance despite being instruction-tuned, potentially due to its limited compositional reasoning capabilities.

## 5.4 CURRICULUM LEARNING EXTENDS PERFORMANCE SCALING BOUNDARY.

We investigate three curriculum learning strategies (Parashar et al., 2025) that control the presentation of training samples according to their difficulty as follows. (i) The easy-to-hard sampling strategy follows a deterministic schedule where samples are organized from easy to hard, enabling the model to gradually acquire increasingly complex patterns. (ii) The balance sampling strategy instead performs uniform random sampling across all difficulty levels, ensuring unbiased exposure throughout training. (iii) The Gaussian sampling strategy models the sampling distribution with a bell-shaped curve whose center progressively moves from easier to harder samples, with tunable parameters controlling the spread and transition speed. Here, we investigate whether curriculum learning can effectively extend the training boundary using Stable Diffusion 1.5 as the base model and test on the GenEval Benchmark. We provide a detailed description in Appendix B.

Our experimental results demonstrate that curriculum learning strategies significantly enhance Comp-Gen performance and successfully extend the scaling boundary beyond conventional training approaches. As illustrated in Figure 4, all three curriculum strategies substantially outperform the baseline model, which maintains a static accuracy of 42% regardless of training duration. The balance sampling strategy achieves a peak performance of 52.1% at 420 training steps, representing a 24% relative improvement over the baseline. More remarkably, the easy-to-hard sampling strategy reaches

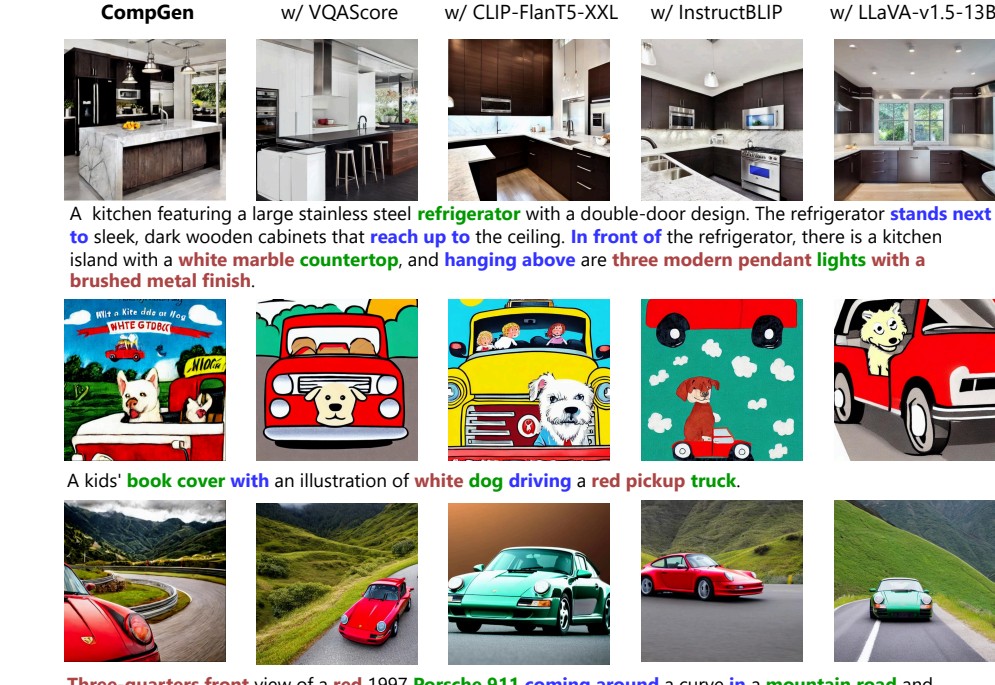

CompGen   w/ VQAScore   w/ CLIP-FlanT5-XXL   w/ InstructBLIP   w/ LLaVA-v1.5-13B

A  kitchen featuring a large stainless steel **refrigerator** with a double-door design. The refrigerator **stands next to** sleek, dark wooden cabinets that **reach up to** the ceiling. **In front of** the refrigerator, there is a kitchen island with a **white marble countertop**, and **hanging above** are **three modern pendant lights with a brushed metal finish**.

A kids' **book cover with** an illustration of **white dog driving** a **red pickup truck**.

**Three-quarters front** view of a **red** 1997 **Porsche 911 coming around** a curve **in** a **mountain road** and **looking over** a **green valley** on a **cloudy** day.

Figure 3: Qualitative comparison of compositional generation across CompGen and ablation models.

53.8% accuracy at 580 steps, while Gaussian sampling attains the highest peak performance of 54.6% at 500 steps — a 30% relative improvement.

Crucially, these results reveal that curriculum learning not only improves absolute performance but also extends the effective training horizon: while traditional approaches plateau early, curriculum strategies continue scaling performance gains well beyond 500 training steps before exhibiting gradual decline. The Gaussian sampling strategy demonstrates the most efficient scaling trajectory, achieving optimal performance with fewer training steps, while easy-to-hard sampling exhibits the most extended scaling boundary, maintaining competitive performance the longest during extended training.

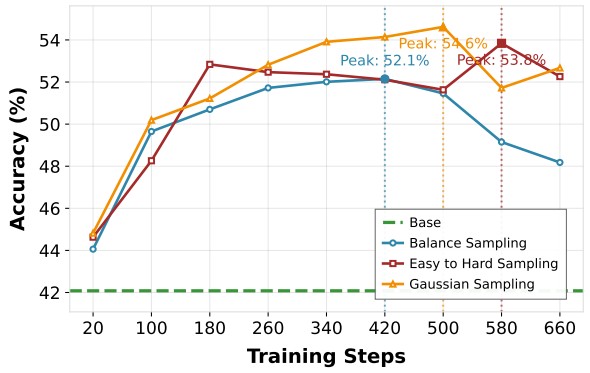

Figure 4: Scaling trend of CompGen with different curriculum learning strategies.

## 6   CONCLUSION

In this work, we introduced CompGen, a novel compositional curriculum reinforcement learning framework that leverages scene graphs and incorporates a principled difficulty criterion with adaptive MCMC sampling, our approach, free from the requirements of need of ground-truth images, systematically improves the ability of T2I models to generate complex scenes with multiple objects, diverse attributes, and intricate spatial-semantic relationships. In the future, it would be interesting to explore more sophisticated metrics that incorporate semantic complexity, visual realism requirements, and cross-modal alignment challenges. Additionally, developing adaptive curriculum strategies that dynamically adjust difficulty based on model performance might further optimize training efficiency.

ETHICS STATEMENT

This work focuses on the study of text-to-image generation and synthesis. All datasets used in our experiments are publicly available and commonly adopted in prior research. We followed the respective dataset licenses and usage terms. No personally identifiable information (PII) or sensitive private data was collected, generated, or annotated by the authors. Our study does not raise direct ethical concerns such as misuse of personal data, harmful content generation, or bias amplification beyond what is already inherent in the benchmark datasets. All generated images in our experiments are for research purposes only and do not involve the creation of deceptive or malicious content.

REPRODUCIBILITY STATEMENT

In order to ensure reproducibility, we provide a comprehensive description of datasets, model implementations, and experimental settings in the main paper and the appendix. The benchmarks and evaluation metrics we used are standard and publicly available. All baselines are either taken from released model checkpoints or trained/evaluated with publicly accessible open-source implementations. To further promote reproducibility, hyperparameters, training details, and evaluation protocol are clearly documented. We release our code at https://anonymous.4open.science/status/T2I-Scaling-D1CC to enable the community to fully reproduce our results. We commit to following ICLR guidelines for transparency and reproducibility in scientific reporting.

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

## A  DATASET

To thoroughly assess the model's compositional capability, we evaluate it on the following five compositional benchmarks: (i) **Geneval** (Ghosh et al., 2023) consists of 553 highly structured prompts spanning six key evaluation dimensions: single object, dual objects, color, count, spatial positioning, and attribute binding. (ii) **T2I-CompBench** (Huang et al., 2025) comprises 6,000 compositional text prompts categorized into three main areas—attribute binding, object relationships, and complex compositions—and further divided into six subcategories: color binding, shape binding, texture binding, spatial relationships, non-spatial relationships, and complex compositions. (iii) **TIFA** (Hu et al., 2023) evaluates generation quality through 4,000 diverse text prompts and 25,000 automatically generated questions using a VQA (Visual Question Answering) model. It spans 12 question categories, including existence verification, object count, color identification, and spatial reasoning. (iii) **DPG-Bench** (Hu et al., 2024) includes 1,065 densely annotated prompts with an average token length of 83.91. These prompts describe complex visual scenarios involving multiple objects and modifiers. (iv) **DSG** (Cho et al., 2023) is an evaluation framework for text-to-image models that generates precise, semantically grounded questions in dependency graphs. DSG ensures reliable evaluations and is supported by the open-sourced DSG-1k benchmark, containing 1,060 diverse prompts.

## B  COMPOSITION CURRICULUM

- **Balance Scheduling.** As a straightforward method to prevent forgetting, the balance scheduling strategy allows for sampling from all tasks with equal probability throughout the entire training process. This can be viewed as a basic form of curriculum learning, or more generally, as the default behavior of most policy optimization algorithms when task difficulty is not considered. For $M$ tasks, the sampling probability $\mathcal{P}_{\text{balanced}}(i, j)$ for task $j$ at any training iteration $i$ is set to:

$$\mathcal{P}_{\text{balanced}}(i, j) = \frac{1}{M}$$

  This means that at each training step, all tasks have an equal chance of being selected. While this method effectively prevents the model from forgetting previously learned tasks, it might introduce more difficult tasks too early, leading to reward sparsity issues and potentially hindering the curriculum learning strategy from achieving optimal results.

- **Easy-to-Hard Scheduling.** Timestep curriculum learning strategies adopt a phased approach, dividing the training process into a series of incrementally difficult stages. In each training phase, the model focuses solely on tasks of a specific difficulty level. We can define the sampling indicator function for task $j$ at training iteration $i$ as $\mathcal{P}_{\text{timestep}}(i, j)$. For a total of $M$ tasks ($j = 1, \ldots, M$), this function takes a value of 1 if the current training iteration $i$ falls within the predefined stage $[\tau_j, \tau_{j+1}]$ for task $j$; otherwise, it is 0. Here, $\tau_j$ denotes the starting training step for the $j$-th stage, with $\tau_1 = 0$ and $\tau_{M+1} = N_T$ being the total number of training steps. This implies that at a given iteration $i$, only tasks corresponding to the current stage are sampled for training.

$$\mathcal{P}_{\text{timestep}}(i, j) = \begin{cases} 1, & \text{if } \tau_j \leq i < \tau_{j+1} \\ 0, & \text{otherwise} \end{cases}$$

  Therefore, at training step $i$, the sampling distribution will be $[\mathcal{P}_{\text{timestep}}(i, 1), \ldots, \mathcal{P}_{\text{timestep}}(i, M)]$.

- **Gaussian Scheduling.** Gaussian scheduling models task sampling as a mixture of Gaussian distributions to provide flexible and fine-grained control over the training process.

  Each task $k$ ($k = 1, \ldots, K$) is assumed to follow a one-dimensional Gaussian distribution with the same variance $\sigma^2$ but different means

$$\mu_k = k - 1.$$

  A latent curriculum position $x_t$ moves from easier to harder tasks as training step $t$ increases:

$$x_t = \left(\frac{t}{T}\right)^{\beta} (K - 1),$$

  where $T$ is the total number of training steps, $\beta > 0$ controls the moving speed of $x_t$, and $K$ is the number of tasks.

The unnormalized sampling score for task $k$ at step $t$ is

$$\mathbf{S}_{\text{Gaussian}}(t, k) = \exp\left(-\frac{(x_t - \mu_k)^2}{2\sigma^2}\right),$$

where $\sigma$ determines the sampling concentration.

The normalized sampling probability is

$$\mathcal{P}_{\text{Gaussian}}(t, k) = \frac{\mathbf{S}_{\text{Gaussian}}(t, k)}{\displaystyle\sum_{m=1}^{K} \mathbf{S}_{\text{Gaussian}}(t, m)}.$$

Smaller $\sigma$ produces sharper, stage-like transitions, while larger $\sigma$ smooths the task shifts. A lower $\beta$ slows the move toward harder tasks, allowing more training on easy tasks in the early phase.

## C   USE OF LLMS

During the preparation of this manuscript, we made limited use of publicly available large language models (LLMs) to assist with English writing. All technical content, including the formulation of ideas, design of methodologies, implementation of experiments, and interpretation of results, was entirely conceived and written by the authors without the involvement of LLMs. The role of LLMs was strictly confined to stylistic and linguistic improvements, in a manner comparable to grammar- or spell-checking software. We ensured that no novel research insights, data, or analyses were generated by LLMs, and all scientific claims and results presented in this work remain the sole responsibility of the authors.

