# OpenReview forum: "Expanding Scaling Boundary of Compositional Text-to-Image Generation via Composition Curriculum"
_ICLR.cc/2026/Conference — ICLR 2026 Conference Withdrawn Submission_

### Official Review · Reviewer_S1uN · 2025-10-28

**Soundness:** 2
**Presentation:** 3
**Contribution:** 3
**Rating:** 4
**Confidence:** 4

**Summary:**

This paper proposes CompGen, a compositional curriculum reinforcement learning framework for text-to-image (T2I) generation. The method leverages scene graphs to measure compositional difficulty and combines adaptive MCMC sampling, fine-grained reward design, and curriculum learning strategies to improve model performance on complex multi-object scenes. Experiments on multiple compositional generation benchmarks demonstrate that CompGen enhances the compositional generation capabilities of both diffusion and autoregressive T2I models.

**Strengths:**

1. The proposed method incorporates multiple well-designed components and demonstrates a certain level of novelty.

2. Experiments validate the fundamental effectiveness of the method.

3. The paper is well-structured and clearly presented.

**Weaknesses:**

1. Clarity of Method: The description of the proposed method is not very clear. Many abstract formulas are used in the paper, which makes it difficult for readers to intuitively understand the approach.

2. Necessity of curriculum learning Data: The paper spends considerable effort on constructing data with varying difficulty levels, but lacks convincing experiments to demonstrate why this stratification is necessary.

3. Need for Additional Experiments: While the paper compares different curriculum learning strategies, it does not explore alternatives such as non-curriculum approaches or randomly constructed data. Including such comparisons could strengthen the validity of the claims.

4. Choice of Base Models: The experiments use relatively small base models (SD1.5 and LlamaGen) with lower baseline performance, which may limit the generalizability and impact of the results.

**Questions:**

1. The experiments use relatively small base models (SD1.5 and LlamaGen) with lower baseline performance. It would be interesting to see whether the method generalizes to stronger baseline models.

2. The motivation for constructing data with varying difficulty levels is not very clear. The paper does not provide convincing results showing why difficulty stratification is necessary. Would simply generating a large amount of training data with an LLM and applying reinforcement learning directly be more effective?

3. Impact of Extreme Difficulty Data: Could the authors discuss the effect of training using only very easy or very hard data, instead of easy-to-hard or mixed curricula? This might provide stronger evidence for the necessity of difficulty-based data stratification.

4. MCMC Sampling Efficiency: Can the authors provide efficiency metrics for the MCMC sampling procedure? How does this construction method compare to simply generating sentences of varying complexity randomly and then filtering them?

---

### Official Review · Reviewer_ipHm · 2025-10-31

**Soundness:** 2
**Presentation:** 3
**Contribution:** 2
**Rating:** 4
**Confidence:** 3

**Summary:**

This paper addresses the challenge of compositional scene synthesis in text-to-image (T2I) generation by proposing the CompGen framework. The approach quantifies scene complexity via scene graphs and uses adaptive MCMC sampling to generate data with controllable difficulty. It integrates curriculum-based reinforcement learning with GRPO optimization, thereby improving the compositional generation capability of both diffusion-based and autoregressive T2I models without modifying the model architecture or requiring labeled images. Experimental results on multiple benchmarks demonstrate notable performance gains.

However, the proposed method has only been evaluated on relatively weak T2I models, making its effectiveness on more mainstream, state-of-the-art models unclear. Furthermore, I believe the paper lacks key ablation studies.

**Strengths:**

1. The overall approach is reasonable and conceptually sound.
2. Effectiveness has been verified on several T2I models.
3. The writing quality is solid.

**Weaknesses:**

1. While the integration of existing approaches such as GRPO and curriculum learning is well executed, the paper’s level of technical novelty may be somewhat limited compared to works that propose entirely new methods.
2. The baseline models are weak, and the set of comparative methods is limited, making it difficult to judge effectiveness convincingly.
3. Missing ablation studies for the curriculum learning component.
4. Visualization results suggest a slight degradation in image quality.

**Questions:**

If you can resolve the following concerns, I am willing to consider raising my score:
1. Could you provide a baseline comparison where GRPO optimization is applied directly without difficulty-based partitioning? I believe this is a critical baseline.
2. From visualization results (e.g., Figure 2’s “squirrel” example), I noticed certain artifacts, which cause concern that your method might degrade image quality. Could you provide quantitative metrics to evaluate visual quality?
3. Could you test your method on stronger, more recent T2I models? As SD1.5 is rather outdated, I suggest validating your approach on advanced models such as Janus-Pro-7B [1] or FLUX.1-dev [2].


[1] Janus-Pro: Unified multimodal understanding and generation with data and model scaling

[2] Black Forest Labs — Flux

---

### Official Review · Reviewer_Jjjp · 2025-11-01

**Soundness:** 2
**Presentation:** 3
**Contribution:** 2
**Rating:** 4
**Confidence:** 3

**Summary:**

This paper proposes CompGen, a novel curriculum reinforcement learning (RL) framework for compositional text-to-image (T2I) generation. The core contribution is a data-centric method that systematically generates training prompts of varying complexity using scene graphs and a novel difficulty metric, then uses these in a GRPO-based RL loop with fine-grained VQA rewards to enhance the compositional capabilities of T2I models. The authors demonstrate performance improvements on multiple benchmarks for both diffusion and autoregressive models and show that curriculum learning strategies effectively extend performance scaling boundaries.

**Strengths:**

1. Novel and Well-Motivated Approach: The idea of using a principled, scene-graph-based difficulty measure to create a curriculum for RL training is novel, well-motivated by human cognitive development, demonstrating effectiveness in enhancing the compositional generation capabilities of T2I models.
2. The paper provides extensive experiments across five established compositional benchmarks and two model architectures (diffusion and autoregressive), demonstrating the generality of the approach.
3. The ablations on reward design and the reward model provide valuable insights into the importance of fine-grained rewards and the choice of the vision-language model (VLM).
4. The investigation into different curriculum schedulers (Easy-to-Hard, Gaussian) is valuable, showing that they not only improve performance but also extend the effective training horizon.

**Weaknesses:**

1. The paper lacks details regarding prompt construction. It does not analyze the vocabulary, syntactic structures, or diversity of the generated prompts. For instance, information about which words are used to build the prompts and the distribution of these words across sampled prompts would be valuable. This absence of transparency makes it difficult to evaluate potential biases in the prompts.
2. The paper lacks discussion of the computational resources and time required for the RL training process. Given that RL for diffusion models is expensive and unstable, this is critical in evaluating the practicality and cost-effectiveness of the method.
3. the paper lacks comparisons with other training-based methods and training-free methods mentioned in the related work (e.g., Attend-and-Excite, RPG-DiffusionMaster) applied to the same base model. This makes it difficult to assess CompGen's relative advantage against the broader field of compositional T2I solutions.
4. The rationale for selecting LLaVA-v1.6-13B over other powerful open-source (e.g., Qwen2-VL) or closed-source (e.g., GPT-4o) VLMs is not provided. An experiment with a top-tier VLM would better indicate the upper bound of performance achievable with this framework.
5. Figure 1 is included in the main text but is not described or referenced

**Questions:**

Please see weaknesses

---

### Official Review · Reviewer_AGhF · 2025-11-01

**Soundness:** 2
**Presentation:** 3
**Contribution:** 2
**Rating:** 4
**Confidence:** 4

**Summary:**

This paper targets at compositional T2I generation, proposing CompGen, a novel compositional curriculum reinforcement learning framework for text-to-image (T2I) generation. It leverage scene graphs and introduces a novel difficulty criterion along with a adaptive Markov Chain Monte Carlo graph sampling algorithm.

**Strengths:**

1. The paper proposes a novel reinforcement learning framework for compositional T2I generation.
2. The paper is well-written and easy to follow.
3. The experiments are extensive.

**Weaknesses:**

1. The motivation of adopting scene graph is not clearly in the paper.
2. The paper is based on the initial difficulty definition. However, the definition may not be suitable (see "Questions" part).
3. There are many other types of information in compositional generation (e.g. action, counting), however the paper seems to only include three types (objects, attributes, and relations).
4. Table 1 shows the performance of the proposed method. However, each benchmark has more fine-grained metrics that were not reported in the paper.

**Questions:**

1. What are the advantages of adopting scene graph generation, compared with just prompt analysis?
2. The definition of difficulty is by three key factors (number of objects, attribute density, relation). However, the difficulties inherent in these three factors for generation models are not the same. Using the multiplier to calculate the difficulty may not be suitable for measuring difficulty.
3. What are the metrics in Table 1 (the average or the sub-metrics)?

---

### Note · Authors · 2025-11-14

I have read and agree with the venue's withdrawal policy on behalf of myself and my co-authors.